# IL-20 Cytokines Are Involved in Epithelial Lesions Associated with Virus-Induced COPD Exacerbation in Mice

**DOI:** 10.3390/biomedicines9121838

**Published:** 2021-12-05

**Authors:** Mélina Le Roux, Anaïs Ollivier, Gwenola Kervoaze, Timothé Beke, Laurent Gillet, Muriel Pichavant, Philippe Gosset

**Affiliations:** 1CIIL-Center for Infection and Immunity of Lille, CHRU Lille, Institute Pasteur de Lille, University Lille, CNRS UMR9017, Inserm U1019, 59000 Lille, France; melina.le-roux@pasteur-lille.fr (M.L.R.); anais.ollivier@pasteur-lille.fr (A.O.); gwenola.kervoaze@pasteur-lille.fr (G.K.); timotheebeke@gmail.com (T.B.); muriel.pichavant@pasteur-lille.fr (M.P.); 2Immunology-Vaccinology Laboratory, Department of Infection and Parasitic Diseases, FARAH, University of Liege, 4000 Liege, Belgium; L.gillet@ulg.ac.be

**Keywords:** COPD, cigarette, viral infection, IL-20 cytokines, lung

## Abstract

(1) Background: viral infections are a frequent cause of chronic obstructive pulmonary disease (COPD) exacerbations, which are responsible for disease progression and mortality. Previous reports showed that IL-20 cytokines facilitate bacterial lung infection, but their production and their role in COPD and viral infection has not yet been investigated. (2) Methods: C57BL/6 WT and IL-20 Rb KO mice were chronically exposed to air or cigarette smoke (CS) to mimic COPD. Cytokine production, antiviral response, inflammation and tissue damages were analyzed after PVM infection. (3) Results: CS exposure was associated with an increase in viral burden and antiviral response. PVM infection in CS mice enhanced IFN-γ, inflammation and tissue damage compared to Air mice. PVM infection and CS exposure induced, in an additive manner, IL-20 cytokines expression and the deletion of IL-20 Rb subunit decreased the expression of interferon-stimulated genes and the production of IFN-λ2/3, without an impact on PVM replication. Epithelial cell damages and inflammation were also reduced in IL-20 Rb^-/-^ mice, and this was associated with reduced lung permeability and the maintenance of intercellular junctions. (4) Conclusions: PVM infection and CS exposure additively upregulates the IL-20 pathway, leading to the promotion of epithelial damages. Our data in our model of viral exacerbation of COPD identify IL-20 cytokine as a potential therapeutic target.

## 1. Introduction

Chronic obstructive pulmonary disease (COPD) remains a major cause of morbidity and mortality worldwide. According to the WHO, COPD is the third leading cause of death worldwide, causing 3.23 million deaths in 2019. Characterized by progressive and irreversible airflow limitation, the main risk factor to develop COPD is cigarette smoking. Chronic exposure to cigarette smoke (CS) triggers inflammatory processes that alters lung barrier functions and reduce immune defense mechanisms, leading to increased susceptibility to respiratory infections with both bacteria and viruses [1]. Such infections further alter the clinical status of COPD patients and are responsible for acute exacerbation episodes. These episodes trigger a vicious circle by favoring the development of a new exacerbation and decrease survival rates in patients with acute exacerbations of COPD (AE-COPD), from 80% to 30% over a five-year period [2]. Although AE-COPD are more frequently associated with bacterial infections, virus infections with Influenza A virus (IAV), respiratory syncytial virus (RSV) and rhinovirus (RV) are also frequent as well as bacteria plus virus co-infections.

The outcome of the infection is tightly related to the efficiency of the anti-microbial host response. Both innate and adaptive immune responses participate in the control of viral infection [3,4]. The immune response against viruses mostly implicates the production of interferon (IFN) response. Whereas the anti-viral effect of type II IFN (IFN-γ) is limited, type I (α and β) and type III (λ or IL-28/IL-29) induced the transcription of several hundreds of genes with potential anti-viral activities [5]. In addition, Th17-type cytokines, including interleukin (IL)-22, are also produced during infection with IAV and RSV. This cytokine controls the inflammatory reaction and preserves the lung homeostasis by limiting tissue injury, particularly through the IL-22 ability to maintain the airway epithelial barrier [6,7]. Moreover, IL-17 and IL-22 are essential in order to prevent secondary bacterial invasion. Indeed, these cytokines orchestrate the anti-bacterial response against different bacteria by modulating the secretion of antimicrobial peptides [8]. IL-22 with IL-19, IL-20, IL-24 and IL-26 belong to the IL-20 cytokine subfamily. Whereas IL-22 acts though a specific IL-10Rb-dependent receptor, IL-19, IL-20 and IL-24 all bind the type I IL-20 receptor (IL-20R), a heterodimeric receptor composed of the IL-20Ra and IL-20Rb chains. Moreover, IL-20 and IL-24 bind the type II IL-20R composed of the IL-22 receptor-α1 subunit and IL-20Rb [9]. Interestingly, these receptors are strongly expressed in airway epithelial cells (AEC) [10], and blocking antibodies against IL-20Rb subunit can efficiently neutralize the response to IL-20 cytokines (IL-19, IL-20 and IL-24) [10] and modulate cutaneous and lung bacterial infection [10,11]. Because IL-20 cytokines are associated with an increased clinical outcome of COPD [12] and viral infection [13,14], we want to assess their physiopathologic role during viral exacerbation of COPD using WT and IL-20Rb KO mice. We analyzed airway inflammation after infection with pneumonia virus of mice (PVM; family *Paramyxoviridae*), a natural mouse pathogen mimicking RSV pathogenesis [15,16]. In contrast to RSV when used in rodent infection models, PVM undergoes robust replication in mouse AEC in vivo [15]. In this study, we demonstrated, for the first time, that PVM infection induces, in an additive manner with CS exposure, the production of IL-20 cytokines, mainly IL-24, and that these cytokines participate in both the inflammatory reaction and the development of lung epithelial lesions.

## 2. Materials and Methods

### 2.1. Animals

Six-to-eight-week-old male wild-type C57BL/6 (H\2Db) mice were purchased from Janvier (SOPF animal facility, Le Genest-St.Isle, France). IL-20Rb KO mice were generated by the group of Wegenka UM [17] and were a generous gift of Pr J Niess (Bern, Switzerland). They were expanded in our animal facility after back cross with C57BL6/J mice.

Mice were exposed to CS (5 cig/day, 5 days/week) during 12 weeks (whole body exposition chamber; Emka, Scireq, Montreal, QC, Canada), to induce COPD-like disease [18]. The control group was exposed to ambient air. All animal work conformed to the guidelines of Animal Care and Use Committee from Nord Pas-de-Calais (agreement no. AF 16/20,090). Mice were maintained in a temperature-controlled (23 °C) facility with a strict 12 h light/dark cycle with food and water provided ad libitum.

### 2.2. Viral Challenge

For infection, frozen working stocks were diluted in phosphate-buffered saline (PBS). Intranasal (i.n) infection was performed under xylazine (16 mg/kg) ketamine (80 mg/kg) anesthesia with 200 PFU of PVM per mouse. Control mice received 50 μL of PBS.

### 2.3. Sample Collection and Processing

Mice were euthanized 9 or 12 days after infection. Bronchoalveolar lavage (BAL) fluids, lungs, spleens and blood samples were collected and kept on ice until they were processed, or frozen immediately in liquid nitrogen. BAL was performed by instilling 0,5 mL (final volume 2,5 mL) of sterile PBS five times. After centrifugation at 400× *g* for 6 min at 4 °C, the supernatant (cell-free BAL fluid) was stored at −20 °C for cytokine analysis (ELISA), and the cell pellet was used for flow-cytometry analysis.

The left lobe of the lung was mashed with a sterile blade and then digested with collagenase (Collagenase Type VI 17104–019 Gibco by Life technologies, Carlsbad, Canifornia United States) at 37 °C. After 15 min of digestion, lungs were homogenized with an 18 G needle and further digested for 15 min. After centrifugation at 400× *g* for 6 min at 4 °C, the pellets were resuspended in a 30% Percoll solution (Percoll TM GE Healthcare 17–0891-01, Chicago, IL, United States) and centrifuged at 500× *g* for 15 min. Total spleen cells were also isolated from crushed spleen and centrifuged at 400 *g* for 6 min at 4 °C. The lung and spleen pellets were resuspended in red blood cells (RBC) lysis buffer during 5 min at room temperature, to remove erythrocytes. The reaction of RBC lysis was stopped with PBS 2% FBS (Gibco by Life technologies, Carlsbad, Canifornia United States). After centrifugation at 400× *g* for 6 min at 4 °C, pulmonary and spleen cells were resuspended in PBS 2% FBS, then enumerated and used for flow cytometry.

### 2.4. Flow Cytometry

BAL, lung and spleen total cells were incubated with the appropriate panel of antibodies for 30 min in PBS 2% FCS. Conjugated antibodies were used against mouse CD5 (ref130–102–574, FITC-conjugated), PBS57-loaded CD1 d Tetramer (NIH facility, PE-conjugated), NK1.1 (ref 130–103–963, PerCp-Cy5.5–conjugated), CD4 (ref 130–102–411, PE-Cy7-conjugated), CD25 (ref 130–102–550, APC-conjugated), CD69 (ref 561–238, Alexa700-conjugated), TCRγδ (ref 130–104–016, APC-Vio770 conjugated), TCR-β (ref 130–104–815, V450-conjugated), CD8 (ref 130–109–252, V500-conjugated), CD45 (ref BLE103140, BV605-conjugated), I-Ab (ref 130–102–168, FITC-conjugated), F4/80 (ref 130–102–422, PE conjugated), CD103 (ref 563–637, PerCP-Cy5.5-conjugated), CD11c (ref 558–079, PE Cy7-conjugated), CD86 (ref 560–581, Alexa-700 conjugated), Ly6G (ref 560–600, APC-H7 conjugated), CD11b (ref 560–455, V45O conjugated), CD45 (ref 130–402–512, V500 conjugated), Ly6C (ref BLE128036, BV605-conjugated) (BD Biosciences, Franklin Lakes, United States; Biolegend, San Diego, United States and Myltenyi Biotech, Paris, France) and CCR2 (ref FAB 5538A, R&D systems, APC conjugated). Data were acquired on a LSR Fortessa (BD Biosciences, Franklin Lakes, United States) and analyzed with FlowJo™ software v7.6.5 (Stanford, CA, USA). Gating strategy has been previously described [8,19]. Absolute cell numbers were calculated according to the total cell number and the frequency of CD45+ immune cells.

### 2.5. Lung Histology

For histopathological analysis, posterior lobes of lungs were fixed with paraformaldehyde (PFA 4%). Lungs were paraffin-embedded and lung section was stained with hematoxylin-eosin. Lung injury was scored based on remodeling and inflammation, as previously defined [8]. Lesions in bronchial epithelium were also evaluated as cell desquamation, cellular lesions and dissociation. The percentage of altered epithelium was determined by using Image J software (NIH).

For immunohischemistry (IHC), lung paraffin-embedded section was deparrafined with two successive baths of Safesolv (Labonord ref 00699464) and rehydrated with successive bath of ethanol (two bath of 100%, one of 90%, one of 80% and one of 50%) and one of water (during 5 min for each). The masking epitope was carried out in pH 6.0 citrate buffer for 15 min at 90 °C. We used as primary antibodies an anti-IL-19 antibody [EPNCIR168] ab154187 (Abcam), an anti-IL-20 antibody (orb13501, Biorbyt, Cambridge, United Kingdom) and an anti-IL-24 antibody (orb228807, Biorbyt, Cambridge, United Kingdom). To quantify the epithelial damages, we used rabbit anti-β-catenin (ref 71–2700), anti-E-cadherin (ref 61–7300) and anti-ZO-1 (ref PA5–19479) antibodies (Thermo Fischer Scientific, Waltham, MA, United States). The UtraTek HRP anti-polyvalent Lab Pack (Histoline laboratories, Pantigliate, Italia) was used according to the manufacturer’s recommendation. Counterstaining was performed with hematoxylin (Interchim, Montluçon, France).

### 2.6. Cytokine Quantification by ELISA

Levels of IFN-γ, CXCL-1, IFN-λ2/3, IL-20, IL-24 (R&D systems, Minneapolis, MN, United States) and IL-19 (Invitrogen, Waltham, MA, United States) were determined in BAL, lung and serum by enzyme-linked immunosorbent assay (ELISA) using the manufacturer’s recommendation.

### 2.7. mRNA Expression Quantification by Reverse Transcription-Polymerase Chain Reaction (RT-qPCR)

Total RNA was isolated from mouse lung using NucleoZol reagent (Macherey-Nagel, Hoerdt, France) according to manufacturer’s instruction. Reverse transcription (RT) was performed with High-Capacity cDNA Reverse Transcription Kit (Applied Biosystems, Waltham, MA, United States) according to manufacturer’s recommendation.

Quantitative PCR was performed to quantify mRNA of interest with QuantStudio 12 K Flex (Applied Biosystems, Waltham, MA, United States) using SYBRGreen (eBiosciences, Waltham, MA, United States) reagent for all gene except for PVM detection using TaqMan^®^ Universal PCR Master Mix (Applied Biosystems, Waltham, MA, United States). The primers (Eurofins Genomics, Ebersberg, Germany) used in this study are described in Table 1.

Results were expressed as mean ± standard error of the mean (sem) of the relative gene expression calculated for each experiment in folds (2−ΔΔCt) using *Hprt1* as house-keeping gene.

### 2.8. Statistical Analysis

The data are expressed as mean ± sem. Results were statistically analyzed with prism software (GraphPad version 9, Prism, San Diego, Ca, USA) using the two-way ANOVA analysis. Virus effect was analyzed by Tukey’s multiple comparison test, while CS or IL-20 KO effect was analyzed by Sidak’s multiple comparison test.

Differences were considered significant when *p* < 0·05, *p* = non-significant (ns). Significant virus effect (in comparison to non-infected mice) is symbolized by * *p* < 0.05; ** *p* < 0.01; *** *p* < 0.001; **** *p* < 0.0001. Significant CS or IL-20^-/-^ effect are symbolized by # *p* < 0.05, ## *p* < 0.01, ### *p* < 0.001 and #### *p* < 0.0001.

## 3. Results

### 3.1. PVM Infection Exacerbates CS-Induced Inflammation and Antiviral Response

An experimental model of COPD exacerbation was established in mice chronically exposed to CS using PVM as the trigger. The effect of an intranasal challenge with PVM was analyzed 9 or 12 days post infection (dpi) (Figure 1a). In this model, PVM RNA was detected in the lung of Air mice with a peak at 9 dpi, and a significant increased detection of PVM RNA was observed in CS-exposed mice at the same time. PVM replication was associated with the induction of antiviral response in the lung at 9 dpi both in Air and CS-exposed mice. An up-regulation of IFN-stimulated gene (ISG) expression, namely, *rig -I, irf-7* and *rsad-2* mRNA expression, was observed in the lung tissues in Air mice with a higher level in CS-exposed mice (except for *irf-7*). Moreover, this was associated with higher IFN-λ2/3 production after PVM infection in CS-exposed mice. In summary, higher PVM replication resulted in an increased antiviral response in CS-exposed mice (Figure 1b).

Histological examination of lung tissues showed that PVM infection induced cell recruitment both in the peribronchial and alveolar area in infected mice at 9 dpi with a peak at 12 dpi. PVM infection also induced significant vascular and bronchial epithelial cell damages at 12 dpi in Air-mice. In contrast, CS exposure induced vascular remodeling in non-infected mice and showed a significant effect at 9 dpi on blood vessels and epithelial damages both in bronchi and alveoli (Figure 1d and as illustrated on Figure 1c). Similarly, epithelial cell lesions were of higher intensity in CS-infected mice at 12 dpi (Figure 1d).

PVM infection was associated with an increased recruitment into the BAL and in cells isolated from lung tissues at 9 and 12 dpi (Figure 2). CS-exposure did not affect total cell recruitment in the pulmonary compartment (10 days after the last exposure to CS). CS exposure increased NKT-like cell recruitment but caused no change in T lymphocytes or cDC2 in the lung. After infection, infiltrated cells included NK cells, NKT-like cells, CD4+ and CD8+ T lymphocytes and dendritic cells (cDC2), in BAL of Air mice at 12 dpi. PVM infection also induced the recruitment of NK cells and cDC2 at 9 dpi and cDC2 and of T lymphocytes at 12 dpi in the lung. CS exposure was responsible for a decrease in NK cell recruitment at 9 dpi into the lung. In the BAL, PVM infection increased the recruitment of NK cells, T lymphocytes and cDC2, as described in Bosteels et al. [20]. NK cell response to PVM was different between Air and CS mice. PVM infection also increased inflammatory mediators such as IFN-γ and IL-1β in BAL fluid, mostly at 9 dpi (Figure 2). IFN-γ production was also induced in the lung at 9 dpi. No effect of CS exposure was observed on these parameters.

At a systemic level, in spleen, no major changes were observed (Appendix A) except for the number of NKT-like cells that was decreased after CS exposure. IFN-γ levels were significantly higher in the serum of infected mice and majored after exposure to CS, whereas IL-1β was not induced after PVM infection.

Altogether, these data showed that PVM replication is more important in CS-exposed mice, leading to a more pronounced lesions within the lung and more inflammation. However, this is not associated with major alterations in the nature of the cell recruitment.

### 3.2. PVM Infection Modulates IL-20 Pathway

Since IL-20 cytokines are known to be involved in bacterial clearance [10,11], we checked on their involvement in the context of a viral infection. We then analyzed the mRNA expression of IL-20 cytokines in the lung. CS exposure tended to increase IL-19, IL-20 and IL-24 mRNA expression in lung (Figure 3a). Moreover, PVM infection also tended to increase the mRNA expression of the 3 cytokines in Air mice at 9 dpi, whereas it had an additive effect with CS exposure on IL-20 and IL-24. In parallel, mRNA levels of IL-20Ra and IL-20Rb subunits were induced by PVM infection in lung only at 9 dpi. CS exposure did not significantly affect IL-20Ra and IL-20Rb subunit expression.

Protein expression was then analyzed by immunohistochemistry on lung sections. IL-19, IL-20 and IL-24 were detected in controls both in Air and CS-exposed mice. However, CS exposure tended to increase IL-19 expression mostly in AEC, whereas staining for IL-20 and IL-24 was more intense in peribronchial infiltrate and in alveolar walls. At 9 dpi after PVM infection in Air mice, a decrease in the staining within the epithelial cells was observed for the three cytokines. In contrast, higher IL-20 and IL-24 expression was observed in peribronchial inflammatory infiltrate and alveolar walls but not for IL-19. In addition, CS-infected mice were characterized by high cell infiltration, which expressed IL-19, IL-20 and IL-24. To confirm these data, protein levels of IL-20 cytokines were also analyzed in BAL fluid (Figure 3b). In the BAL, the secretion of the three cytokines was increased by PVM infection at day 9 for IL-19 and IL-24 and at 12 dpi only for IL-20 (Figure 3c). CS exposure did not alter their concentrations in the BAL. In the sera, no significant changes were observed for IL-19 and IL-24. Blood IL-20 levels were transiently increased with a maximum at 9 dpi without changes in CS-exposed mice.

Altogether, these data showed that infection by PVM increased the expression and the secretion of IL-20 cytokines with a different timing according to the cytokine and the compartment. Our data also confirm that AEC are an important source of these cytokines. Exposure to CS mostly affected their expression within the lung.

### 3.3. IL-20 Cytokines Have a Deletorious Effect during PVM Infection in CS-Exposed Mice

To evaluate the role of IL-20 cytokines during viral exacerbation of COPD, we exposed IL-20Rb KO mice and WT mice to CS and PVM using the same protocol as described in Figure 1a. As expected, PVM mRNA was detected in the lung of CS WT mice and the IL-20Rb depletion did not affect the PVM mRNA expression in the lung (Figure 4a). Concerning the antiviral response in the pulmonary compartment, *rig-I* and *rsad-2,* mRNA lung expression and IFN-λ2/3 production in the BAL were reduced in IL-20Rb KO in comparison to WT mice (Figure 4a), whereas *irf-7* was not modulated (data not shown).

Histological analysis revealed that there was no difference in WT and IL-20Rb KO mice in not infected CS-exposed mice. In contrast, this analysis confirmed that PVM infection was associated with a strong inflammation in CS WT mice 12 dpi. Moreover, PVM infection both at 9 and 12 dpi was associated with epithelial and vascular damages in CS-exposed WT mice. Interestingly, the inflammatory infiltrate within the peribronchial and alveolar area, and the tissue lesions, was markedly reduced in infected IL-20Rb KO mice both at 9 and 12 dpi (Figure 4b). This was confirmed by a decrease in the histologic score and the surface of the epithelial lesions (Figure 4b), whereas the vascular damages were not modulated (data not shown).

Inflammatory cell recruitment was measured in the BAL and the lung tissues (Figure 5a). Total cell number was lower in the BAL fluid of IL-20Rb KO mice in comparison with WT mice. However, no significant modulation of lymphocyte or cDC2 counts was observed in KO mice compared to WT mice, whereas the number of NK cells and NKT-like cells tended to be lower in KO mice. In the lung compartment, total cell number as well as NK, NKT-like and T lymphocyte recruitment were not modified in IL-20Rb KO mice. Immunomodulatory cytokine analysis showed an increase of IFN-γ level in the BAL and the lung tissues at 9 dpi in WT mice, whereas IL-1β concentrations were not significantly modulated (Figure 5b). In IL-20Rb KO infected mice, IFN-γ level was decreased in the lung but not in the BAL. In contrast, IL-1β level was not significantly modulated by the deletion of IL-20Rb

In the spleen, CS exposure was associated with an increased number of NK and NKT cells in IL-20Rb KO mice. In the blood, the PVM-induced IFN-γ secretion was decreased in IL-20Rb KO mice whereas IL-1β concentrations were unchanged (Appendix A).

These data reveal that IL-20 cytokines play a deleterious role in the virus-induced exacerbation of COPD by modulating the inflammation and the tissue lesions, a process associated with decrease in ISG expression and interferon production.

### 3.4. Implication of IL-20 Cytokines during PVM Infection in Control Mice

To determine if the role of IL-20 cytokines was similar in control mice, we performed the same protocol in Air-exposed mice. As expected, infection by PVM induced a strong anti-viral response with a peak at 9dpi associated with inflammatory reaction and lung tissue lesions (Appendix A). The deletion of IL-20Rb receptor in Air-exposed mice amplified the replication of PVM and the expression of *Rsad-*2 only at 9dpi but not of *rig-I* and IFN-2/3, whereas the levels were similar at 12dpi for these three targets. Regarding pulmonary lesions, we did not observe significant effect on the histologic score, a result associated with the lack of effect on the cell infiltrate and the cytokine production in the lung (Appendix A). Nevertheless, blocking the IL-20Rb also significantly decreased the epithelial lesions in Air mice at 12 dpi (Appendix A).

Altogether, these data showed that the deletion of IL-20Rb transiently increased the replication of PVM in Air mice without altering the inflammatory reaction. As observed in CS-exposed mice, it also decreased epithelial lesions at 12 dpi.

### 3.5. IL-20 Cytokines Play a Role in Lung Permeability during PVM Infection

Since IL-20Rb KO mice show fewer epithelial damages, we evaluated the lung permeability by measuring the protein concentration in BAL fluid in CS-exposed mice. As expected, PVM infection in WT mice strongly amplified the protein concentration in the BAL both at 9 and 12 dpi. Interestingly, IL-20-Rb deletion was associated with lower levels of protein in infected mice but not in non-infected mice (Figure 6a). After PVM infection, both WT and IL-20Rb KO mice lose weight (Figure 6b). Only IL-20Rb KO mice had regained weight at 12 dpi in CS-exposed mice. In Air mice, PVM infection also increased the protein concentration in BAL and induced some weight loss (Appendix A), similar to in CS-exposed mice. Blocking the IL-20 pathway did not affect these parameters.

To confirm these data, we analyzed, by immunohistochemistry, the expression of junction proteins, including E-cadherin (Figure 6c), β-catenin and Zonula Occludens-1 (ZO-1) (data not shown). As suspected, PVM infection was associated with a disorganization of junction proteins both in bronchial and alveolar epithelia, as illustrated by E-cadherin staining at 9 dpi. In infected WT mice, the staining was more diffuse and not limited to the apical side of the bronchial epithelial cells, and these alterations were more pronounced in CS-exposed mice (Figure 6c) compared with Air mice (Appendix A). The staining was also present in inflammatory cells. In contrast, PVM infection in IL-20Rb KO mice strongly reduced these alterations both in Air and CS-exposed mice at 12 dpi.

Altogether, these data revealed that IL-20 overexpression was associated with an increased lung permeability in response to PVM infection, a process associated with intercellular junction disorganization.

## 4. Discussion

An exacerbation is a striking event in COPD patient, which marks the transition from stability to a rapid decline of lung function [21]. Infection by virus, including RSV, is one of the main factors responsible for COPD exacerbation [22]. Due to the consequences of infection on the progression of the disease [23], it is critical to establish the mechanism by which RSV-induced exacerbation leads to a decline in lung function [24]. Since RSV is not a natural pathogen for mice, we have used PVM, which infects mice and induces a pathology very close to RSV in human [16] to study the mechanism in AE-COPD. In our model mimicking COPD, PVM challenge resulted in a greater virus proliferation associated with enhanced inflammation and epithelial damage in the lungs. Antiviral elements were mainly induced at 9 dpi when the viral load was maximal in the lungs, and the inflammatory response was observed later at 12 dpi. This study is the first one that examines the impact of PVM infection in mice chronically exposed to CS and demonstrates a pathogenic role for IL-20 cytokines.

Antiviral response such as interferon is critical in limiting viral replication [25]. COPD is associated with a decreased antiviral pathway leading to an increased susceptibly to viral infection [26,27,28,29]. At steady state, COPD patients show a downregulation of their antiviral components, including a decrease in interferon stimulated-gene and IFN-β expression, especially in the more severe state of the disease [27,30]. In addition, it has been demonstrated that CS extract can directly dampen the response to IAV from human PBMC, as depicted by a reduced production of IFN-β, IFN-γ and RIG-I expression [29]. These findings seem to be different according to the viral trigger that is used. Indeed, in our COPD mimicking model exacerbated by PVM, we observed a significant increase in the antiviral response. This increase in antiviral pathway, including RIG-I, Rsad-2 and IFN-λ2/3, was linked to an increased PVM detection in the lung of CS-exposed mice. This was consistent with other studies demonstrating the increased expression of antiviral elements such as MDA-5, IFN-β and IFN-λ1, by bronchial epithelial cells from COPD patient stimulated by rhinovirus or RSV [31,32] and leading to an increased viral burden in mice that, in turn, might be responsible for the increased anti-viral response [33]. Similarly to RSV, PVM infection enhanced inflammation associated with perivascular and bronchial damages in CS-exposed mice [33]. The infiltration of immune cells following PVM infection mainly consists of the recruitment of macrophages, neutrophils, NK, lymphocytes and cDC2 [34,35,36] and was associated with a majored production of IFN-γ in CS-exposed mice, as observed in RSV exacerbated COPD [33].

IL-20 cytokines are associated with many inflammatory diseases, such as psoriasis, rheumatoid arthritis, COPD or infectious diseases [9]. In our study, chronic exposure to CS and PVM infection are able to induce IL-19, IL-20 and IL-24 expression in structure cells and inflammatory cells in the lung. Similarly, IL-24 is also induced by viruses such as IAV or HIV [37,38]. In our study, we observed that CS exposure and viral expression had an additive effect on the expression of these cytokines, a process that we have previously reported for bacterial infection with *Streptococcus pneumoniae* [10]. Most research has been focused on IL-22, but IL-20 cytokines can induce cell recruitment through the production of chemokines and can modulate the production of antiviral molecules and cytokines [9]. IL-24 supplementation is responsible for increased expression in antiviral elements, including *mxa, oas* and *ifn-**β,* in A549 AEC, whereas its inhibition decreases interferon response and viral burden [37]. We report here in the PVM context that IL-20Rb depletion reduced production of *rsad-2* and of IFN-λ2/3 in CS-exposed mice, whereas it enhanced *rsad-2* expression in Air-exposed mice. In parallel, we detected more PVM replication in Air-exposed mice, whereas it was unchanged in CS-exposed mice. The difference might be related to the conjunction of different factors with opposite effects in CS-exposed mice. Indeed, CS exposure alters the anti-viral response and activates the IL-20 pathway leading to the upregulation of the IL-20 cytokines and the IL-20 cytokine receptor subunits, namely, IL-20Ra and IL-20Rb in infected mice. These data suggest that the role of IL-20 cytokines on anti-viral response is dependent on the type of virus and probably of the degree of IL-20 pathway activation. This convergence might also explain why IL-20Rb depletion reduced the lung inflammation and epithelial injury in CS-exposed mice while only decreasing the epithelial lesions in Air-mice. In CS-exposed mice, the effect on inflammation was linked to a decreased number of inflammatory cells within the BAL without modification of their repartition. This suggests that IL-20 cytokines affect a major factor involved in leucocyte recruitment rather than a specific component responsible for the mobilization of leucocyte subtype.

IL-20Rb KO mice exhibited better outcomes of the disease in CS-exposed mice. We previously demonstrated that blocking IL-20Rb pathway promotes bacterial clearance [10], and this effect is associated with a lower histopathological score measuring both the inflammation and the airway remodeling. In this model, the impact on PVM infection is probably related to the decreased alteration of AEC, associated with a better preservation of intercellular junction expression and lung permeability. In Air-exposed mice, the depletion of IL-20Rb only reduced epithelial lesions. We can hypothesize that the effect on lung permeability requires a high level of either IL-20 cytokines or IL-20 receptors. Since lung permeability is identified as a marker of severity, these data suggest that blocking the IL-20 pathway can improve the outcome of this disease. At this stage, we can formulate two hypotheses in order to explain this role for IL-20 cytokines: (1) these cytokines participate in the virus-induced lesions in AEC. We have performed some in vitro experiments that did not allow us to show a direct effect of IL-20 cytokine supplementation on in vitro response of AEC in terms of cytotoxicity and anti-viral response; (2) our data also suggest that IL-20 cytokine expression might be associated to a loss of epithelial barrier function, particularly when the epithelium is affected by another environmental factor such as CS exposure. Hsu et al. demonstrated that recombinant IL-20 treatment decreased E-cadherin expression in prostate epithelial cells [39]. Additional experiments are required in order to analyze the epithelial permeability in a model of air–liquid interface after viral infection and the role of IL-20 cytokines. Moreover, it has been shown that some polymorphism in IL-19 and IL-20 genes could worsen the outcome of chronic hepatitis B infection [14] and septic shock [40]. IL-20 cytokines may be a common link between these diseases. Therefore, targeting these cytokines and/or their receptors could represent a good approach to treat COPD and some infectious diseases. Therefore, we need to better understand the mechanisms underlying the beneficial effects of the treatment using anti-IL-20Rb antibody on tissue damages, and we need to check if anti-IL-20Rb antibody could limit pulmonary viral infections not only with RSV but also with another important cause of AE-COPD, the influenza A virus.

In conclusion, we herein report for the first time that PVM infection exacerbates CS-induced inflammation. PVM infection modulates the IL-20 pathway and interacts with smoking. Our data identify the IL-20 cytokine pathway as a potential therapeutic target, particularly in the context of viral exacerbation of COPD, due to its ability to limit the epithelial lesions.

## Figures and Tables

**Figure 1 biomedicines-09-01838-f001:**
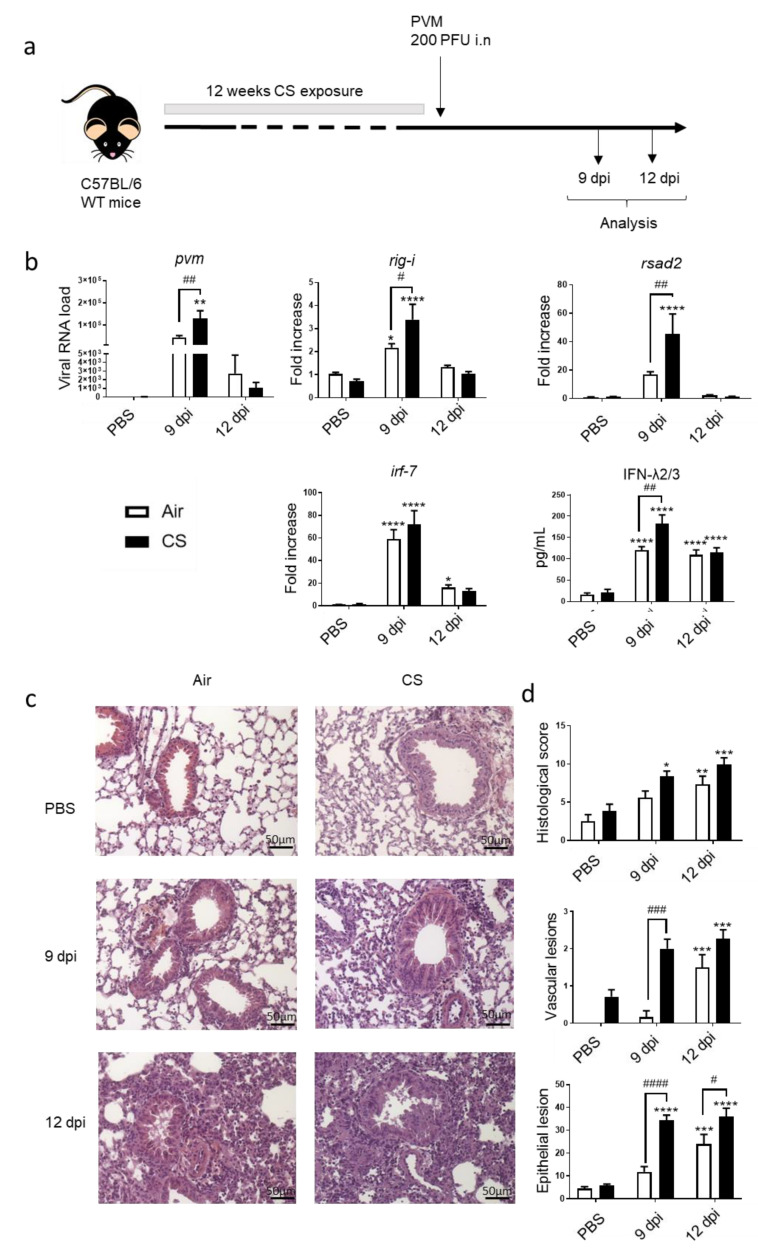
PVM infection exacerbates COPD. (**a**) Experimental design of the study: the day of the infection was defined as day 0, allowing the association of this time point with the end of cigarette smoke (CS) protocol, consisting of mouse exposure to ambient air (Air) or CS (5 cigarettes/day, 5 days week) for 12 weeks. Analysis was performed at 9 and 12 days post infection (dpi). (**b**) Viral load and antiviral response, including mRNA expression rig-i, rsad-2 and irf-7, was evaluated by RT-qPCR in lung tissues. Results were expressed as fold increase compared to Air mice exposed to PBS using hprt1 expression as a house keeping gene. IFN-λ2/3 was evaluated by ELISA (pg/mL). (**c**) Histological changes were evaluated at 9 and 12 dpi. Representative slides are shown after HE staining. Scale bar = 50 µm. (**d**) Histological score analysis including vascular lesion and epithelial damages is expressed as mean ± SEM. Air mice (white bars) and CS mice (Black bars). * *p* < 0.05, ** *p* < 0.01, *** *p* < 0.001 and **** *p* < *0*.0001 correspond to virus effect (PVM vs. PBS). # *p* < 0.05, ## *p* < 0.01, ### *p* < 0.001 and #### *p* < 0.0001 correspond to CS effect (CS vs. Air). Three independent experiments have been performed with 3–5 mice in each group per experiment.

**Figure 2 biomedicines-09-01838-f002:**
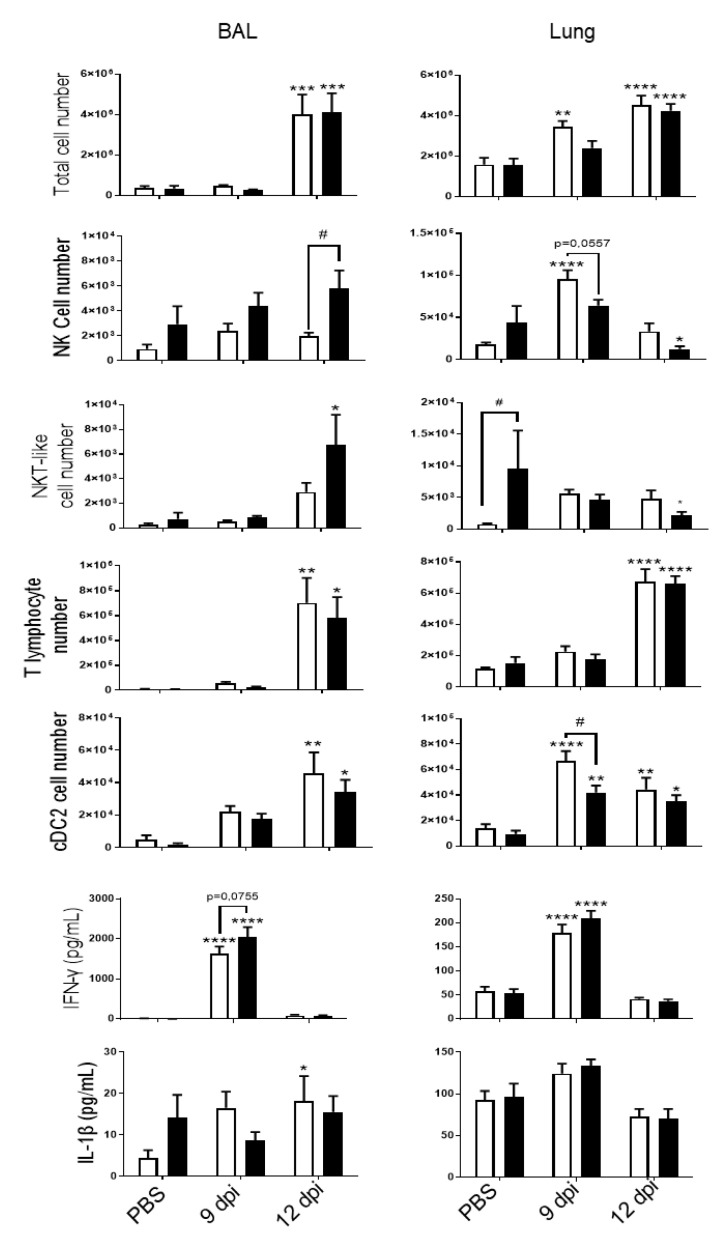
PVM infection induces lung inflammation. Total cell numbers, CD45 +/TCRβ-/NK1.1+ NK cells, CD45+/TCRβ+/NK1.1+ NKT-like cells, CD45+/TCRβ+ CD4+ and CD8+ T lymphocytes and CD45+/ F4/80-/CD11c+/CD11b+/CD103-dendritic cells (cDC2) count were evaluated in BAL fluid (**left**) and lung tissue (**right**) of Air (**white bars**) and CS mice (**Black bars**). IFN-γ and IL-1β levels were analyzed by ELISA in BAL fluid (**left**) and lung tissue (**right**) of Air and CS mice. Data represent the mean ± SEM. * *p* < 0.05, ** *p* < 0.01, *** *p* < 0.001 and **** *p* < 0.0001 correspond to virus effect (PVM vs PBS). # *p* < 0.05 corresponds to CS effect (CS vs. Air). Three independent experiments have been performed with 3–5 mice in each group per experiment.

**Figure 3 biomedicines-09-01838-f003:**
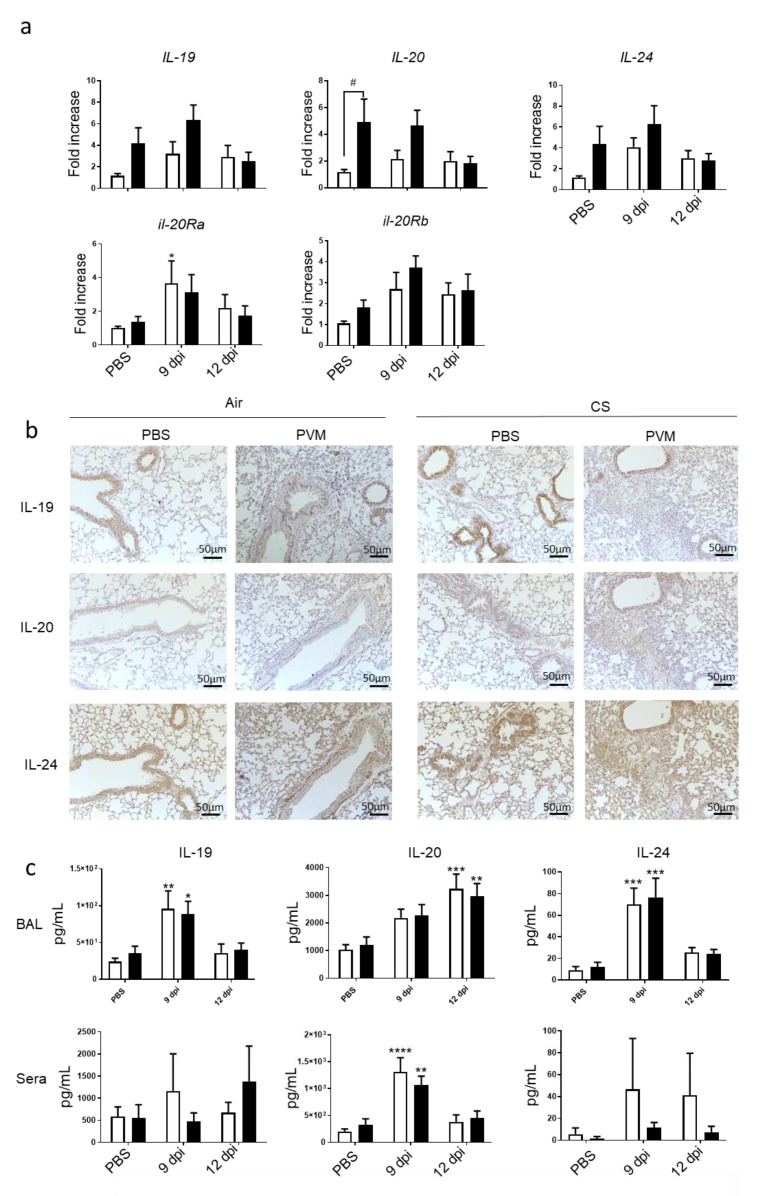
PVM infection modulates IL-20 cytokines. (**a**) Il-19, *Il-20*, *Il-24, Il-20ra* and Il-20rb mRNA levels were evaluated by RT-qPCR in lung tissue at 9 and 12 dpi. Results were expressed as fold increase compared to Air mice exposed to PBS and using expression of hprt1 as a housekeeping gene. (**b**) Expression of IL-19, IL-20 and IL-24 was evaluated on lung sections by immunohistochemistry at 12 dpi in Air and CS mice. Scale bar = 50 µm. (**c**) *Il-19*, *Il-20*, and *Il-24* protein level in BAL fluid and sera were analyzed by ELISA and expressed in pg/mL. Results were expressed as mean ± SEM. * *p* < 0.05, ** *p* < 0.01, *** *p* < 0.001 and **** *p* < 0.0001 correspond to virus effect (PVM vs PBS). # *p* < 0.05 corresponds to CS effect (CS vs. Air). Three independent experiments have been performed with 3–5 mice in each group per experiment.

**Figure 4 biomedicines-09-01838-f004:**
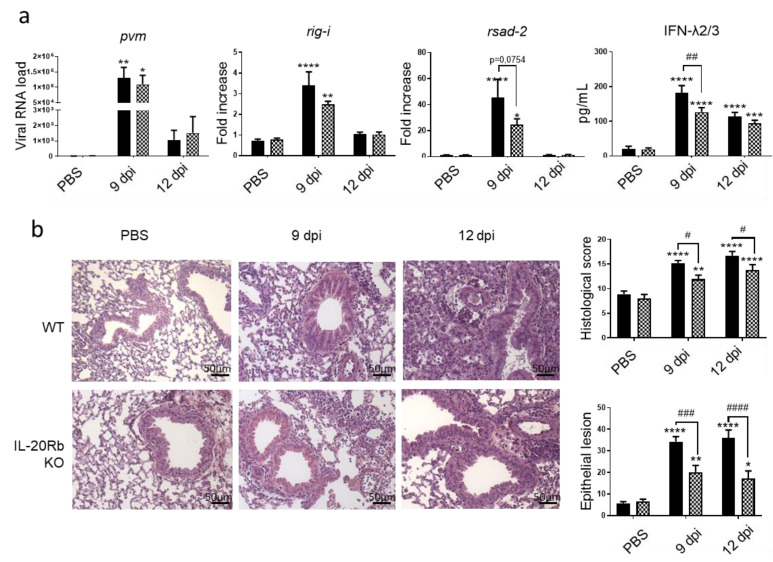
PVM infection induces a lower response in IL-20Rb KO mice exposed to CS. (**a**) Viral load and antiviral response, including mRNA expression of *rig-I* and *rsad-2*, evaluated by RT-qPCR in lung tissues. Results were expressed as fold increase compared to Air mice exposed to PBS using hprt1 expression as a houskeeping gene. IFN-λ2/3 was evaluated by ELISA (pg/mL). (**b**) Histological changes were evaluated at 9 and 12 dpi. Scale bar = 50 µm. Histological score analysis, including epithelial damages, is expressed as mean ± SEM. WT mice (**Black bars**) and IL-20Rb KO mice (**Hashed bars**). * *p* < 0.05, ** *p* < 0.01, *** *p* < 0.001 and **** *p* < 0.0001 correspond to virus effect (PVM vs. PBS). # *p* < 0.05, ## *p* < 0.01, ### *p* < 0.001 and #### *p* < 0.0001 correspond to IL-20Rb KO effect (IL-20Rb KO vs. WT). Three independent experiments have been performed with 3–5 mice in each group per experiment.

**Figure 5 biomedicines-09-01838-f005:**
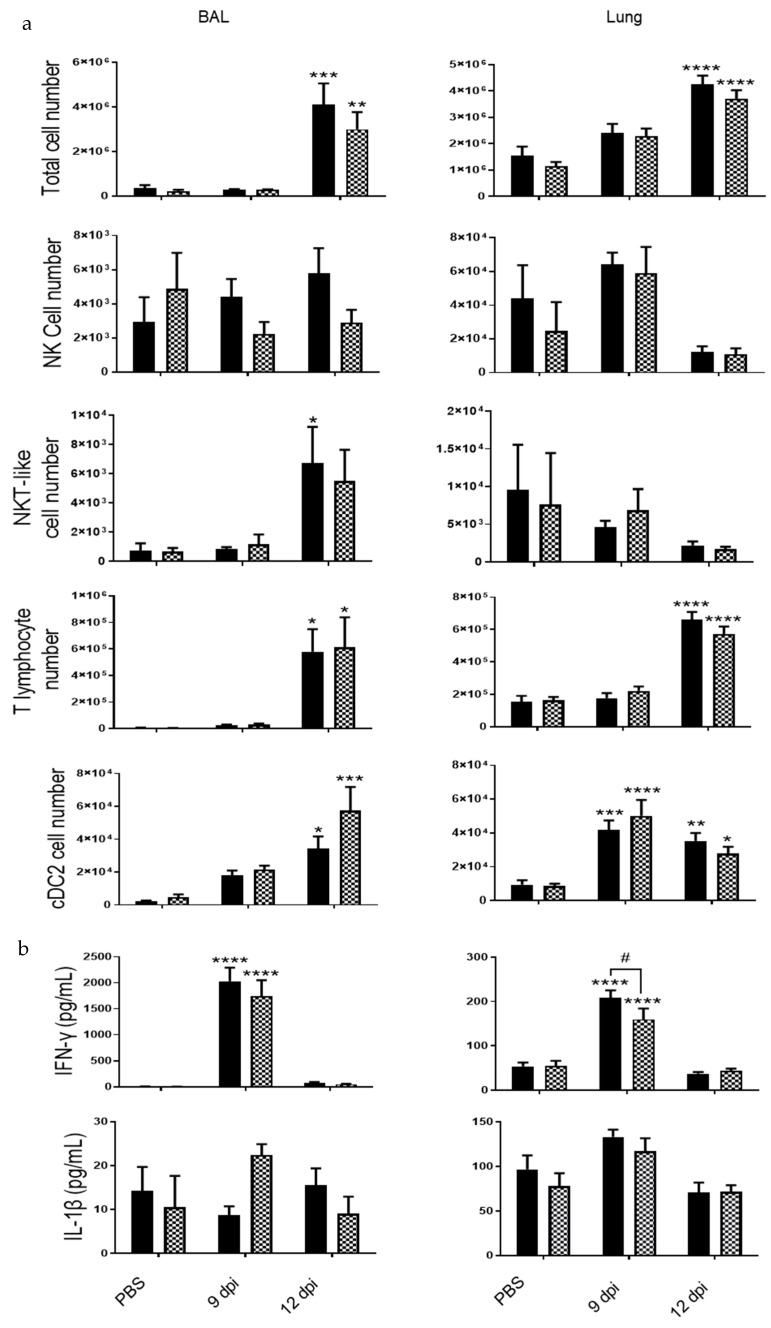
PVM infection induces lung inflammation both in WT and IL-20 KO mice. (**a**) Total cell number, CD45+/TCRβ-/NK1.1+ NK cells, CD45+/TCRβ+/NK1.1+ NKT-like cells, CD45+/TCRβ+ CD4+ and CD8+ T lymphocytes and CD45+/ F4/80-/ CD11c+/CD11b+/CD103- dendritic cells (cDC2) count were analyzed in BAL fluid and lung tissue of WT (**white bars**) and IL-20Rb KO mice (**hashed bars**). (**b**) IFN-γ and IL-1β levels were analyzed by ELISA (pg/mL). Data represent the mean ± SEM. * *p*< 0.05, ** *p* < 0.01, *** *p* < 0.001 and **** *p* < 0.0001 correspond to virus effect (PVM vs PBS). # *p* < 0.05 corresponds to IL-20Rb KO effect (IL-20Rb KO vs. WT). Three independent experiments have been performed with 3–5 mice in each group per experiment.

**Figure 6 biomedicines-09-01838-f006:**
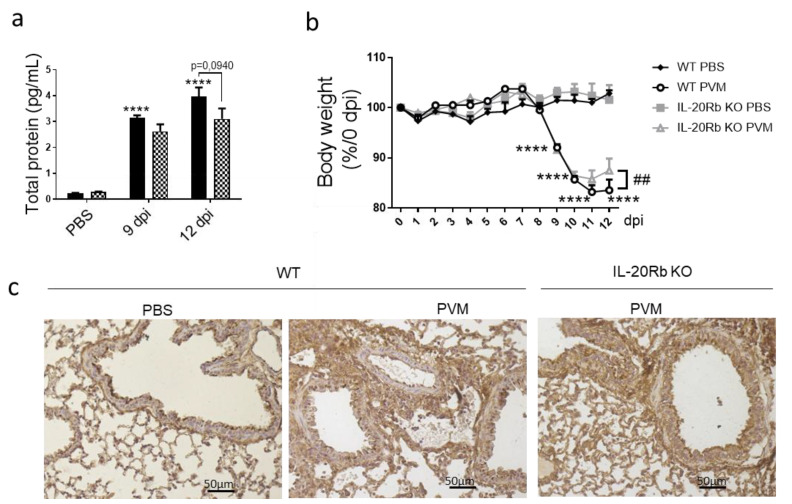
PVM infection modulates lung permeability and E-cadherin protein expression. (**a**) Lung permeability was analyzed through the quantification of total protein in BAL fluid (pg/mL) in smoking WT mice (**Black bars**) and IL-20Rb KO mice (**Hashed bars**). (**b**) Body weight follow up expressed as percentage of body weight at 0 dpi. (**c**) Expression of E-cadherin was evaluated on lung sections by immunohistochemistry at 12 dpi in smoking WT mice and IL-20Rb KO mice. Scale bar = 50 µm. Data represent the mean ± SEM. **** *p* < 0.0001 correspond to virus effect (PVM vs. PBS). Three independent experiments have been performed with 3–5 mice in each group per experiment.

**Table 1 biomedicines-09-01838-t001:** Sequence of primers used for qPCR.

Gene	Sequences
*hprt1*	Sens: 5′ TCC TCC TCA GAC CGC TTT T 3′ Antisens: 5′ CCT GGT TCA TCA TCG CTA ATC 3′
*il-19*	Sens: 5′ TGT GTG CTG CAT GAC CAA CAA 3′ Antisens: 5′ GGC AAT GCT GCT GAT TCT CCT 3′
*il-20*	Sens: 5′ TCT TGC CTT TGG ACT GTT CTC 3′ Antisens: 5′ GTT TGC AGT AAT CAC ACA GCT TC 3′
*il-24*	Sens: 5′ AGC ACT GGC CCT TTC TTC AA 3′ Antisens: 5′ TGG CAA GAC CCA AAT CGG AA 3′
*rig-i*	Sens: 5′ TGC GGA AAT ACA ACG ATG CA 3′ Antisens: 5′ GCT CGG TCT CAT CGA ATG CTG 3′
*rsad-2*	Sens: 5′ TGC TGG CTG AGA ATA GCA TTA GG 3′ Antisens: 5′ GCT GAG TGC TGT TCC CAT CT 3′
*pvm*	Sens: 5′ GCC GTC ATC AAC ACA GTG TGT 3′ Antisens: 5′ GCC TGA TGT GGC AGT GCT T 3′ Probe: 5′[FAM] C GCT GAT AAT GGC CTG AG CA [TAM] 3′

## Data Availability

Data available on request due to restrictions. The data presented in this study are available on request from the corresponding author. The data are not publicly available due to the potential therapeutic developments.

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
