# Peer review of "IL-20 Cytokines Are Involved in Epithelial Lesions Associated with Virus-Induced COPD Exacerbation in Mice"

_biomedicines, 2021, doi:10.3390/biomedicines9121838_

Round 1

Reviewer 1 Report

In their study entitled "IL-20 cytokines are involved in epithelia lesions associated with virus-induced COPD exacerbation in mice" the authors demonstrated that PVM infection induces in an additive manner with CS exposure the production of IL-20 cytokines, mainly IL-24, and that these cytokines participate in both the inflammatory reaction and the development of lung epithelial lesions.

The study is worthy, well designed, well performed and well written. I have only minor comments.

Minor comments:

  1. Introduction section, line 1: "Chronic obstructive pulmonary disease" should precede "COPD".
  2. Introduction section, last paragraph, lines 3-4: "production interferon" should be "production of interferon". 
  3. At the end of the Introduction section: "In this study, , we first, demonstrated" is more proper to be "In this study we demonstrated, for the first time".
  4. Introduction section, line before the last one: "participate to" is more proper to be "participate in".
  5. Lung histology section, page 3, line 3: "Lung injury were scored" should be "Lung injury was scored".
  6. Discussion section, first paragraph, line 7: "EA-COPD" should be "AE-COPD".
  7. Discussion section, second paragraph, line 16: "mice" should be added after "CS-exposed".
  8. Discussion section, third paragraph, line 8: "can induced" should be "can Induce".
  9. In many locations through the manuscript comma (,) should be added between the words.
  10.   

Author Response

In their study entitled "IL-20 cytokines are involved in epithelia lesions associated with virus-induced COPD exacerbation in mice" the authors demonstrated that PVM infection induces in an additive manner with CS exposure the production of IL-20 cytokines, mainly IL-24, and that these cytokines participate in both the inflammatory reaction and the development of lung epithelial lesions.

The study is worthy, well designed, well performed and well written. I have only minor comments.

We would like to warmly thank the reviewer for these comments.

Minor comments:

  1. Introduction section, line 1: "Chronic obstructive pulmonary disease" should precede "COPD".
  2. Introduction section, last paragraph, lines 3-4: "production interferon" should be "production of interferon". 
  3. At the end of the Introduction section: "In this study, , we first, demonstrated" is more proper to be "In this study we demonstrated, for the first time".
  4. Introduction section, line before the last one: "participate to" is more proper to be "participate in".
  5. Lung histology section, page 3, line 3: "Lung injury were scored" should be "Lung injury was scored".
  6. Discussion section, first paragraph, line 7: "EA-COPD" should be "AE-COPD".
  7. Discussion section, second paragraph, line 16: "mice" should be added after "CS-exposed".
  8. Discussion section, third paragraph, line 8: "can induced" should be "can Induce".
  9. In many locations through the manuscript comma (,) should be added between the words.

We have changed the manuscript according to the comments. Please see the marked version of our article.

Reviewer 2 Report

The authors have shown that IL-20-related cytokines adversely affect the acute exacerbation of COPD by viral infection using IL-10Rb-deficient mice. However, there are problems in drawing this conclusion.

Major point

1) In order to investigate the effects of IL-20-related cytokines, authors conducted experiments using IL-10Rb-deficient mice. However, they compared PMV-induced changes in IL-10Rb-deficient mice with in wild-type mice, only in the CS system in which cigarette smoke was inhaled. In order to investigate the effect of IL-20-related cytokines on the acute exacerbation of COPD by PVM infection, it is necessary to compare it in system that does not induce COPD. Otherwise, they may only be looking at the function of IL-20-related cytokines against PVM infection. Wild-type and IL-10Rb-deficient mice should be compared to PVM infection under air conditions and histological and in vivo data should be shown in addition to Figure 6.

2) There are IL-19, IL-20, and IL-24 as IL-20-related cytokines. If it is possible to show which of these cytokines has the strongest effect, for example, by using cytokine specific neutralizing antibodies, it seems that the understanding of the pathological condition will be better.

Minor point

1) For each experiment, the number of experiments and the number of samples for statistical analysis should be specified.

Author Response

The authors have shown that IL-20-related cytokines adversely affect the acute exacerbation of COPD by viral infection using IL-10Rb-deficient mice. However, there are problems in drawing this conclusion.

We would like to warmly thank the reviewer for the comments and suggestions to improve the quality of our manuscript.

Major point

1) In order to investigate the effects of IL-20-related cytokines, authors conducted experiments using IL-10Rb-deficient mice. However, they compared PMV-induced changes in IL-10Rb-deficient mice with in wild-type mice, only in the CS system in which cigarette smoke was inhaled. In order to investigate the effect of IL-20-related cytokines on the acute exacerbation of COPD by PVM infection, it is necessary to compare it in system that does not induce COPD. Otherwise, they may only be looking at the function of IL-20-related cytokines against PVM infection. Wild-type and IL-10Rb-deficient mice should be compared to PVM infection under air conditions and histological and in vivo data should be shown in addition to Figure 6.

We have included some control mice (exposed to air) in our experiments as suggested by the reviewer. We did not include these data in the first version of the manuscript in order to simplify the message. As requested, we have now added these data to the revised version of the article. As illustrated in the supplemental figure 3, the deletion of IL-20Rb receptor in Air mice amplifies the replication of PVM and the expression of Rsad2 only at 9dpi whereas the levels are similar at 12dpi. Regarding the tissue lesions in the lung, we did not observe a significant effect on the histologic score, a result associated with the lack of effect on the cell infiltrate and the cytokine production in the lung (supplemental figures 3-4). Nevertheless, blocking the IL-20R also significantly decreases the epithelial lesions in Air mice at 12dpi and also allows to maintain the expression of E-Cadherin in AEC (supplemental figure 5).

Altogether, these data show that the deletion of IL-20Rb transiently increase the replication of PVM in Air mice without modification in the inflammatory reaction. As observed in CS-exposed mice, it also decreases epithelial lesions at 12dpi. The limited effect of this treatment in Air mice is probably related to the lack of modulation of IL-19, IL-20 and IL-24 by PVM infection. We have also commented this point in the discussion.

2) There are IL-19, IL-20, and IL-24 as IL-20-related cytokines. If it is possible to show which of these cytokines has the strongest effect, for example, by using cytokine specific neutralizing antibodies, it seems that the understanding of the pathological condition will be better.

Indeed, it might be interesting to define the physiopathologic role of each of the IL-20 cytokines. However, our experimental model is very time-consuming. For ethical and financial reasons, we cannot evaluate the effect of specific neutralizing antibodies against the 3 cytokines. Moreover our data demonstrate that the 3 cytokines are produced after infection by PVM and might be implicated in the response to this virus (figure 3). All these 3 cytokines are sharing receptor subunits, including the IL-20Rb subunit. For these reasons, we are presently focusing the therapeutic approach on the IL-20 receptors and more specifically on IL-20Rb. Our aim is to produce a neutralizing anti-IL-20Rb antibody that we can propose as a therapeutic tool.                      

Minor point

  • For each experiment, the number of experiments and the number of samples for statistical analysis should be specified.

We have now added the number of experiments and the number of mice per group in each figure legend from the revised version.

Round 2

Reviewer 2 Report

The author has made possible corrections to this paper.

I have nothing more to point out.